# Cardiotoxicity in Acute Myeloid Leukemia in Adults: A Scoping Study

**DOI:** 10.3390/cancers16132474

**Published:** 2024-07-06

**Authors:** Ioannis Konstantinidis, Sophia Tsokkou, Savvas Grigoriadis, Lalayianni Chrysavgi, Eleni Gavriilaki

**Affiliations:** 12nd Propaedeutic Department of Internal Medicine, Ippokratio University Hospital, Department of Medicine, Faculty of Health and Sciences, Aristotle University of Thessaloniki, 54124 Thessaloniki, Greece; ikonsc@auth.gr (I.K.); stsokkou@auth.gr (S.T.); 2Research Team “Histologistas”, Interinstitutional Postgraduate Program “Health and Environmental Factors”, Department of Medicine, Faculty of Health Sciences, Aristotle University of Thessaloniki, 54124 Thessaloniki, Greece; 3Medical School, University of Thessaly, 41334 Larissa, Greece; sgrigoriadis@gmail.com; 4Hematology Department, BMT Unit, G Papanicolaou Hospital, 57010 Thessaloniki, Greece; luizana6@gmail.com

**Keywords:** acute myeloid leukemia (AML), cardiotoxicity, pharmaceutical options, cardiovascular complications, cardio-oncology

## Abstract

**Simple Summary:**

This is a scoping study aiming to extensively assess and explore the degree of cardiotoxicity in patients with Acute Myeloid Leukemia (AML) that can be caused due to pharmaceutical treatments. Many pharmacological regiments used can potentially cause cardiotoxicity in AML patients, but it is understandable that being familiar with all the available treatment options available and every potential adverse effect is impossible before the initiation of the therapy. However, hematologists and, in general, physicians should try to be updated with the most recent information released to improve the quality of life of their patients and minimize the risk of additional complications. Correct communication and collaboration between hematologists and cardiologists is of paramount importance and a huge advantage that can yield promising results, utilizing both cardiology imaging techniques, such as cardiac ultrasound, which is an easy and cost-effective means of instant diagnosis and staging the suspected complication, and serum cardiac biomarkers, according to current cardio-oncology guidelines.

**Abstract:**

**Introduction:** According to the National Cancer Institute of the NIH, acute myeloid leukemia (AML) is a rapidly growing cancer with a large quantity of myeloblasts. AML is most often observed in adults over the age of 35, accounting for 1% of all cancer types. In 2023, the number of new cases being reported was estimated to reach around 20,380 in total and the rate of mortality in the same year was 1.9%, or 11,310 cases, in the US. **Purpose:** This scoping study aims to extensively assess and explore the degree of cardiotoxicity in patients with AML that can be caused due to pharmaceutical treatments prescribed by hematologists. This is achieved by performing extensive searches of different scientific databases including PubMed, Scopus, and ScienceDirect. **Results:** A variety of options are available that are summarized in tables included herein, with each having their advantages and risks of adverse effects, among these being cardiotoxicity. Important medications found to play a significant role include gemtuzumab ozogamicin, venetoclax, and vyxeos. **Conclusions:** It is understandable that being familiar with all the treatment options available and every potential adverse effect is impossible. However, hematologists and, in general, physicians must try to be updated with the most recent information released to improve the quality of life of their patients and minimize the risk of additional complications.

## 1. Introduction

According to the National Cancer Institute of the NIH, acute myeloid leukemia (AML) is a rapidly growing cancer where a large quantity of myeloblasts—immature white blood cells—are present in the blood and bone marrow. Most common in older individuals, AML can spread to various body parts including the lymph nodes, spleen, liver, central nervous system (CNS), skin, gums, and the testicles [1].

### 1.1. Epidemiology of AML

AML is mostly observed in adults over the age of 35, accounting for 1% of all cancer types. In 2023, the number of new cases being reported was estimated to reach around 20,380 in total and the rate of mortality in the same year was 1.9%, or 11,310 cases, in the USA. Additionally, AML is more commonly diagnosed with increasing age and the median age of diagnosis is 69 years. It is more prevalently observed in the male rather than female population, with the rate of new cases being 5.0 and 3.4 per 100,000 people, respectively. The death rate was 2.7 per 100,000 per year, specifically in non-Hispanic white individuals, based on cases from the period 2016–2020. The 5-year relative survival rate was estimated to be 31.7% in adult patients with AML [2]. Additionally, it is interesting to mention the global burden of the disease, as discussed by Ou et al., 2020. Specifically, AML deaths globally have increased by 93% in the period 1990 to 2017, whereas the number of DALYs due to AML increased by 56% during the same period of time [3] (Figure 1).

### 1.2. AML and Subtypes

According to the classification by FAB, although not really used in clinical practice currently, there are nine subtypes of AML, all listed from most common to most rare in Figure 2.

### 1.3. AML Classification

According to the 5th edition of the WHO classification of myeloid neoplasms, AML is divided into two main groups. The first is AML with defining (recurrent) genetic abnormalities, and the second group can be defined by the type of differentiation displayed [4]. All the types are included in Table 1 and Figure 2.

### 1.4. Symptomatology (Figure 3)

The clinical manifestation of AML varies widely from symptoms of anemia, such as fatigue, weakness, shortness of breath during normal activities, lightheadedness, headaches, and pale complexion; thrombocytopenia, for instance, skin bruising, excessive and uncontrolled bleeding, gingiva bleeding, and, for females, frequent menstrual bleeding; leukopenia, and mostly neutropenia, which can lead to frequent infections; and leukostasis, which is a life-threatening complication caused by high concentrations of circulating leukemic cells that cause severe tissue hypoxia [5,6,7].

**Figure 3 cancers-16-02474-f003:**
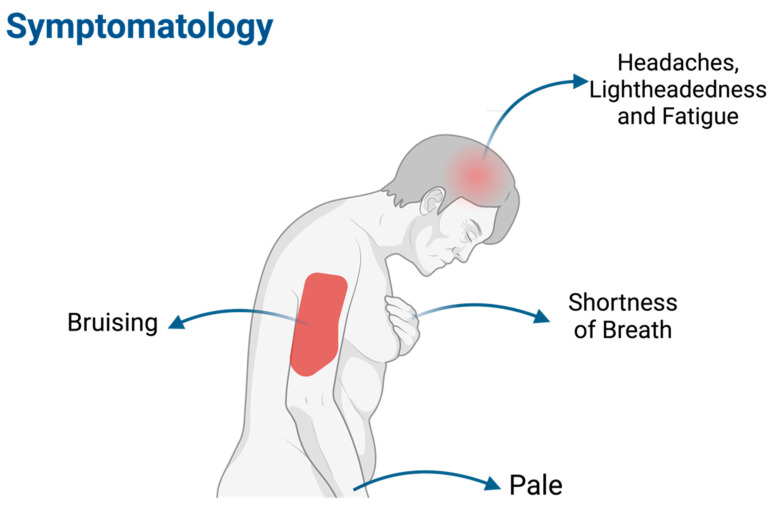
Symptomatology. Created using BioRender [accessed on 20 May 2024). Available online: https://app.biorender.com/illustrations/65ff040d4603630b1370a481].

## 2. Methodology

This scoping study aims to assess and explore the possibility of cardiotoxicity that is caused by pharmaceutical treatments prescribed by hematologists in patients with AML. This was achieved by performing an extensive search of different scientific databases including PubMed, Scopus, ScienceDirect, the American Cancer Society. The most crucial information found is summarized in Table 2 and is further explained in the main text. This scoping study was not registered as a systematic review.

## 3. Results

Table 2 summarizes the findings, including those for systematic chemotherapy drugs and targeted therapy drugs.

### Treatment Regimens in AML in Adults

The pharmaceutical regimens currently used in the treatment of AML in adults are summarized in Table 2.

## 4. Discussion

### 4.1. Systemic Chemotherapy Drugs Used to Treat AML 

#### 4.1.1. Anthracyclines (Daunorubicin, Idarubicin, Mitoxantrone)

Anthracyclines are a class of antibiotics primarily used to treat various forms of cancer. They are derived from certain strains of the Streptomyces bacterium [8]. They function similarly by inserting themselves in between DNA base pairs, which causes the double helix structure of DNA to uncoil. This mechanism inhibits DNA synthesis and modifies DNA topoisomerase II activity. Additionally, anthracyclines might generate free radicals, which can harm quickly proliferating cells [9]. Anthracyclines, such as daunorubicin, idarubicin, and mitoxantrone, have been used for induction chemotherapy in the treatment of adult AML, alone or together with cytarabine analogs in the classical “7 + 3” regimen.

Anthracyclines have been proven to be associated with many adverse effects, such as allergic reactions—anaphylaxis, severe cough, photosensitivity, skin and nail hyperpigmentation—as well as voice hoarseness, flushing face, fatigue, joint pain, painful or difficult urination, and persistent diarrhea. The general term anthracycline-induced cardiotoxicity encompasses acute heart failure, arrhythmias, or myocarditis, and anthracycline-related left ventricular dysfunction [ARLVD] [9,10,11].

#### 4.1.2. Cytarabine

Cytarabine, also known as arabinosylcytosine (ARA-C), is a pyrimidine analog. It is converted into triphosphate within cells and competes with cytidine to incorporate itself into DNA and works as an antimetabolic agent that blocks the function of DNA polymerase. It is FDA-approved for induction therapy in adult AML patients in the classic “7 + 3” regimen together with an anthracycline.

The most feared side effect of cytarabine is myelosuppression, which manifests as significant megaloblastic alterations and acute or severe pancytopenia. Stomatitis, conjunctivitis, dermatitis, gastrointestinal problems, reversible hepatic enzyme increase, and discomfort at the site of injection are among the other, less serious symptoms. Anaphylaxis, a severe form of hypersensitivity, is extremely rare but is a clear sign that cytarabine therapy should be stopped. Cytarabine can be administered intrathecally for CNS treatment or prophylaxis, but it may cause cerebral and cerebellar toxicity. Brain poisoning manifests as convulsions and dementia, whereas cerebellar toxicity manifests as ataxia and slurred speech. Angina pectoris, pericarditis, anthracycline-associated cardiomyopathy, and bradycardia (2.8%) are possible side effects of cytarabine’s potential cardiovascular toxicity. Cytarabine syndrome is an uncommon condition that appears soon after cytarabine is administered. It is treatable with corticosteroids and is characterized by nonspecific symptoms such as fever, malaise, rash, joint pain, and muscular discomfort. Patients with renal impairment are likely to have severe symptoms more frequently as the drug’s renal clearance declines [12].

#### 4.1.3. Hypomethylating Agents (Azacitidine and Decitabine)

Hypomethylating agents including azacitidine and decitabine are cytidine analogs in which the carbon atom at position 5 in the pyrimidine ring has been replaced by a nitrogen atom that causes DNA demethylation by the inhibition of DNA methylotransferase-1 (DNMT-1). Azacitidine has been approved for use as a treatment for adult patients not eligible for hematopoietic stem cell transplantation with AML with 20–30% blasts and multilineage dysplasia, whereas decitabine is used for the treatment of older patients (older than 65 years old) with AML who are not eligible for treatment with standard induction chemotherapy [13].

Treatment with azacitidine has been correlated with atrial fibrillation (3% to 5%) and congestive heart failure (3% to 11%), in addition to pericarditis, pericardial effusion, and a case of sudden cardiac death, whereas decitabine can cause myocarditis and reversible cardiomyopathy (<5%) [14]. Other AEs of both hypomethylating agents are pancytopenia, hepatotoxicity, acute renal injury and, additionally, chest pain and hypo- or hypertension.

#### 4.1.4. CPX-351 (Daunorubicin and Cytarabine)

CPX-351, known as Vyxeos in the United States and in the European Union/United Kingdom as Vyxeos liposomal, was established as a liposomal encapsulation of daunorubicin and cytarabine in a synergistic 1:5 molar ratio. It has been FDA-approved for the treatment of newly diagnosed, therapy-related AML or AML with myelodysplasia-related changes in adults and children aged ≥1 year in the United States and adults aged ≥18 years in Canada, the European Union, and the United Kingdom since 2017 [15]. The lower cardiotoxicity of CPX-351 relative to daunorubicin plus a cytarabine free-drug combination in hiPSC-derived cardiomyocytes in vitro confirmed that the liposomal formulations of anthracyclines assuage anthracycline-induced cardiomyopathy [16]. Other CPX-351 AEs are, apart from anthracycline-associated cardiomyopathy, edema, arrhythmia, hypo- or hypertension, chest pain, GI disturbances (constipation, diarrhea, nausea, vomiting), cytopenias, allergic reactions, and ototoxicity.

### 4.2. Targeted Therapies Used to Treat AML 

#### 4.2.1. Gemtuzumab Ozogamicin

Gemtuzumab ozogamicin (GO; Mylotarg) is a humanized anti-CD33 monoclonal antibody coupled with calicheamicin, an antitumor antibiotic that produces dsDNA disruptions [17]. It is used in the treatment of AML and, specifically, its FDA-approved indications are favorable, intermediate, or unknown cytogenetic groups, patients with CD33+-expressing leukemic blasts and, finally, as a frontline treatment with intensive chemotherapy [18].

GO’s most common adverse effects include myelosuppression, increased hepatic enzyme activity, infections, fever and chills, bleeding, nausea and vomiting, and dyspnea [17,18]. Neutropenia and thrombocytopenia and oral mucositis or stomatitis are found to be the most common [19]. GO-specific AEs are reported to be associated with hepatotoxicity (hyperbilirubinemia and elevated transaminases), with the clinical manifestations of veno-occlusive disease (VOD), as mentioned in Leopold et al. 2002 [19], and portal fibrosis, described by Perry et al., 2005 [20].

Cardiotoxicity due to GO has not been a frequently reported adverse event. Concerning CVAEs, 5% of patients receiving therapy with GO presented with infusion-related hypotension, stated by Leopold et al., 2002 [19], and, in a case report by McNerney et al. 2022, it was reported that a pediatric patient with r/r AML developed acute LV dysfunction following GO monotherapy, suggesting a latent correlation between GO and cardiotoxicity in heavily pre-treated AML patients [21].

#### 4.2.2. Midostaurin

Midostaurin is a multi-kinase inhibitor with activity against both FLT3-ITD and FLT3-TKD (type 1 inhibitor). It is used in the treatment of AML and, specifically, its FDA-approved indications are newly diagnosed FLT3 ITD/TKD-mutated AML and as a frontline regimen with intensive chemotherapy [22].

Midostaurin’s adverse effects include pancytopenia, infection, mucositis or stomatitis, pneumonitis or pulmonary infiltrates, GI tract intolerance (diarrhea, nausea), hepatotoxicity (increased alanine aminotransferase and hyperbilirubinemia), and electrolyte disturbances (hypokalemia, hyponatremia, hypophosphatemia, and hypocalcemia) [23].

Regarding midostaurin’s cardiotoxic effect, Wilson and Pemmaraju, 2021, proved its association with QTc prolongation, peripheral edema, hypotension, risk of heart failure development, and pericardial effusion in 6% of the patients included in their study [24]. Atrial fibrillation is also strongly associated with midostaurin use in AML patients [25,26].

#### 4.2.3. Quizartinib

Quizartinib selectively hinders class III receptor tyrosine kinases, including colony-stimulating factor 1 receptor (CSF1R/FMS), FMS-related tyrosine kinase 3 (FLT3/STK1), stem cell factor receptor (SCFR/KIT), and platelet-derived growth factor receptors (PDGFRs). It is approved in Japan for R/R FLT3-mutated AML.

Quizartinib’s reported adverse effects involve prolonged cytopenias and QTc prolongation, possibly via K+ channel blockage [22]. As reported by Boluda et al., 2023, the occurrence of QTc prolongation in the QuANTUM-First phase 3 clinical trial with quizartinib was 34% in the quizartinib vs. 18% in the control group [27]. Other cardiac adverse effects including dyspnea, cardiomyopathy, and heart failure, which usually occur with multi-targeting TKIs, have been described during treatment with quizartinib as well [28].

#### 4.2.4. Enasidenib

Enasidenib is a selective inhibitor of the mutant IDH2 enzyme, which promotes the generation of 2-hydroxyglutarate and stimulates the differentiation of leukemic blasts. Enasidenib is FDA-approved for the treatment of adults with mutant-IDH2 R/R AML [29].

Enasidenib’s adverse effects comprise cytopenias, GI disturbances, such as nausea, vomiting, and decreased appetite, hepatotoxicity with bilirubin increase, and differentiation syndrome [29]. Enasidenib is also associated with moderate elevations in aminotransferases during therapy and is suspected to trigger rare occasions of clinically manifesting acute liver injury [30].

#### 4.2.5. Gilteritinib

Gilteritinib is an effective type 1 inhibitor of both FLT3-ITD and FLT3-TKD and also has activity against AXL [22]. It is the second FLT3 inhibitor to be approved by the FDA and is used in the treatment of relapsed or refractory FLT3-mutated AML.

Common adverse events of gilteritinib are pancytopenia, hepatotoxicity, GI disturbances, such as diarrhea or constipation and nausea or vomiting, hypokalemia, and infections, mainly pneumonia [31]. Regarding the cardiovascular complications of gilteritinib use, QTc prolongation was present in 5% of participants and peripheral edema was reported as a clinical manifestation of acute heart failure during treatment in a study by Perl et al., 2019 [31].

#### 4.2.6. Glasdegib

Glasdegib is an inhibitor of the Hedgehog signal transduction pathway that binds to Smoothened transmembrane protein (SMO), leading to decreased Glioma Associated Oncogene (GLI) transcription factor activity and downward signaling of the pathway. The Hedgehog signaling pathway is required to maintain the leukemic stem cell population; therefore, glasdegib binding and SMO inhibition reduce AML levels of GLI1 and AML leukemia initiation potential [32]. It is approved by the FDA for use in combination with low-dose cytarabine in newly diagnosed AML patients aged ≥75 years or those who have comorbidities that impede the use of intensive induction chemotherapy [33].

AEs more common during treatment with glasdegib encompass febrile neutropenia, alopecia, muscle spasms, and GI disturbances, such as nausea and vomiting, anorexia, constipation, and dysgeusia [34]. Cardiac adverse effects include peripheral edema (as a manifestation of acute heart failure) and QTc prolongation [33].

#### 4.2.7. Ivosidenib

Ivosidenib is a selective inhibitor of mutant IDH1. Mutant IDH1 converts α-ketoglutaric acid (α-KG) to 2-hydroxyglutarate (2-HG), which inhibits cell differentiation and promotes tumorigenesis in both hematologic and nonhematologic malignancies [35]. Ivosidenib is FDA-approved as a first-line monotherapy for ND IDH1-AML patients if they are ≥75 years or unfit for intensive chemotherapy, a first-line therapy with the IVO/AZA regimen for ND IDH1-AML patients if they are ≥75 years or unfit for intensive chemotherapy, and as monotherapy for relapsed or refractory IDH1-AML.

Differentiation syndrome, QT prolongation, and leukocytosis are considered Ivosidenib-specific AEs. QT prolongation has been seen in 18–25% patients with AML, but none have needed permanent cessation of ivosidenib.

#### 4.2.8. Venetoclax

Venetoclax is a selective monoclonal antibody inhibitor of B-cell lymphoma 2 (BCL-2), a counter-apoptotic protein. The overexpression of BCL-2 has been shown in AML cells, and aids tumor cell survival and has been associated with resistance to chemotherapy. Venetoclax in combination with azacitidine or decitabine is FDA-approved for elderly/unfit patients with AM. It is frequently used both in the upfront and relapsed/refractory setting based on the higher efficacy and improved survival seen with the combination regimen compared with HMA alone [14,36].

The most frequently reported adverse effects of venetoclax are cytopenias (>10%), GI tract disturbances [nausea, constipation, diarrhea, vomiting (>30%)], hypokalemia, pneumonia (38%), sepsis (6%), and tumor lysis syndrome (1%). Cardiotoxicity caused by venetoclax is mainly associated with atrial fibrillation in 5% of AML patients [14,36]. The cardiotoxic effect of venetoclax was confirmed to be around 29% as a percentage of its cardiac AEs attributable to AFib [37].

#### 4.2.9. Olutasidenib

Olutasidenib is a selective, small-molecule inhibitor of the mutated IDH1 enzyme with the therapeutic capability to restore regular cellular differentiation, and has been approved for elderly or unfit patients with AML as a first-line treatment together with azacitidine [38].

The adverse effects of olutasidenib include cytopenias, acute liver injury and failure (ALT, AST, γ-GT), GI side effects, pneumonia, hypokalemia, differentiation syndrome, and tumor lysis syndrome. Cardiotoxicity associated with olutasidenib involves QTc interval prolongation that has been described in 8% patients [38].

### 4.3. Cardiovascular Complications in AML

Since acute leukemia (AL) shares systemic pathogenic pathways and mechanisms with cardiovascular diseases (CVDs), such as inflammation, metabolic changes, clonal hematopoiesis, angiogenesis changes, extracellular matrix, and stroboscopic cells, it may be linked to direct heart injury [39].

#### 4.3.1. Risk of CVDs in Patients with Newly Diagnosed Acute Leukemia in Adults

A population-based real-world retrospective cohort study in China by Xiao et al., 2022 [39], tried to investigate the correlation between newly diagnosed AL in adults with heart-related lesions. They demonstrated that the AL patients had already experienced some heart-related injuries before the diagnosis of AL, accompanied by a significant increase in cardiac enzymes, such as CK-MB, LDH, hs-cTnI, BNP, and LVID, and a decline in the ejection fraction (EF). Thus, they tried to establish a CVD risk marker for primary AL patients and verified that LVID was the most reliable indicator of cardiac injury in AL patients, although age and EF were independent risk factors for the prognosis. A positive correlation was observed between the index of heart injury and the ratio of bone marrow blasts. These findings offer fresh perspectives on the intricate etiology and molecular processes that result in cardiac injury in AL patients, as well as how to notify medical professionals when cardiac injury is present in AL patients. To validate and investigate the precise mechanism underlying the relationship between leukemia and cardiac abnormalities, more research is still required and the invention of a risk score for CVDs in AL patients is of paramount importance [39].

#### 4.3.2. Myocarditis

Leukemic cardiac involvement might present with nonspecific signs and symptoms that vary depending on the area of the heart that has been invaded. As such, the diagnostic procedure may be difficult. Heart failure, pericardial effusion, and, very infrequently, constrictive pericarditis can be symptoms of leukemic infiltration. Acibuca et al., 2019, described such a case of a man that had undergone allogenic peripheral blood stem cell transplantation due to M4 AML and presented with acute systolic dysfunction, significant heart enlargement, and constrictive pericarditis that led to the diagnosis of myocarditis due to leukemic heart infiltration. Thus, in addition to cardiotoxic medications or opportunistic infections, leukemic cardiac involvement should be considered when a patient with a hematologic malignancy presents with heart failure, myocardial hypertrophy, and constrictive pericarditis [40].

#### 4.3.3. Acute Coronary Syndromes (ACSs)

A case of AML resulting in acute coronary artery thrombosis, anterolateral ST elevation myocardial infarction, ventricular tachycardia, and pulseless electrical activity was reported by Ferrel et al., 2022. The electrocardiogram and telemetry findings of this patient show a dramatic progression from a normal sinus rhythm to acute myocardial infarction to a terminal rhythm, indicating an extreme case of acute myocardial infarction with disseminated intravascular coagulopathy causing acute coronary thrombosis and sudden death. When AML and acute myocardial infarction occur simultaneously, the prognosis is dramatic. Systemic coagulopathy, platelet dysfunction, and thrombocytopenia make management difficult, and using thrombolytic drugs can be lethal [41]. Another similar, yet less lethal, case was reported by Jao et al., 2014, when a patient with a medical history of coronary artery disease and MDS presented with intermittent exertional angina and was diagnosed with STEMI as a result of the newly diagnosed AML transformation of the MDS [42]. Thus, it is of the utmost importance that clinicians bear in mind the possibility of AML-derived ACS, as the management of these patients is different and difficult.

#### 4.3.4. Leukostasis

AML induces hyperleukocytosis, which compromises tissue perfusion and results in leukostasis, a medical emergency. It usually affects the brain and lungs; heart involvement is extremely uncommon. A case of AML with acute coronary syndrome due to leukostasis-induced myocardial ischemia was reported by Manogna et al. in 2020. The patient had typical anginal chest pain, ST segment elevations in their ECG, and an increase in cardiac enzymes (high-sensitivity Troponin I). This patient’s case represents 6% of ACS cases caused by leukostasis [43]. Another case by Maldonado et al., 2016, reports another extremely rare clinical manifestation of leukostasis: a patient presenting with high-output cardiac failure in the clinical setting of AL and leukostasis [44].

### 4.4. Cardio-Oncology and Cardiotoxicity

#### 4.4.1. Definitions

Cardio-oncology is an emerging field in cardiology that tries to elucidate the correlation between the heart and cancer, and is distinguished into three categories: cancer treatment-related cardiotoxicity; cardiovascular complications of the disease itself; and cardiac tumors [45]. Cardiotoxicity, on the other hand, is described as the development of cardiac muscle dysfunction, such as a decreased left ventricular ejection fraction, as a result of exposure to antineoplastic therapy, which can lead to heart failure [46].

#### 4.4.2. Cardiovascular Toxicity Risk Stratification Scores

The dynamic course of CTR-CVT development in cancer patients is governed by the assumption that the absolute risk is determined by their baseline risk and changes over time as they are exposed to cardiotoxic medications. The ESC’s Heart Failure Association (HFA) and the International Cardio-Oncology Society (ICOS) collaborated on the publication of the current recommendations on this topic. The aforementioned risk can be defined in two ways: firstly, the likelihood of occurrence and, secondly, the severity of the complication. After cardiotoxic cancer treatment is provided, a novel risk assessment is necessary to determine different long-term CV health trajectories [45].

A pre-treatment CTR-CVT risk assessment should preferably be carried out using a well-known risk stratification method that includes numerous risk indicators to evaluate the patient-specific risk, as furtherly discussed in the current recommendations of the 2022 ESC guidelines on cardio-oncology. It should be mentioned that these risk stratification scores were developed using specific cancer patient groups. Before beginning potentially cardiotoxic anticancer therapy, all cancer patients should undergo CV toxicity risk stratification, which includes obtaining a clinical history, performing a physical examination, an ECG, and a general blood test, and measuring HbA1c, the lipidemic profile, cardiac serum biomarkers, and/or TTE (depending on the cancer drug type and CV toxicity risk) (Class I). Patients who are classified as having a low risk of CV toxicity should begin anticancer treatment right away (Class I). Cardio-oncology referral may be suggested for patients with moderate CV toxicity risk (Class IIb). Cardio-oncology referral is advised in high-risk and very-high-risk patients prior to anticancer therapy (Class I) [47].

However, as previously mentioned, these guidelines and risk stratification scores have not been developed in adult AML patients and, regarding AML-related therapeutic regimens, they only mention anthracycline-related toxicity scores, but they could offer a general principle of the risk that an AML-patient could possibly have.

#### 4.4.3. Cumulative Incidence and Risk Factors of Cardiac Events in AML Patients

Boluda et al., 2023, tried to estimate the cumulative incidence (CI) of cardiac events in newly diagnosed AML patients and to identify risk factors for their occurrence. The CI of life-threatening cardiac events was 4% at 6 months and 6% at 9 years. They occurred more frequently in patients not included in clinical trials and those treated with intensive approaches (although this was not statistically significant). Additionally, patients with prior cardiac antecedents had an increased incidence of fatal cardiac events compared with the patients with no prior cardiac disease. Regarding non-fatal cardiac events, the CI was 43% at 6 months and 57% at 9 years and risk factors, such as an age over 65 years old, relevant cardiac antecedents, and non-intensive chemotherapy as a front-line treatment, were correlated with an increased CI of non-fatal cardiac events [27]. However, more clinical studies need to be carried out to identify more risk factors for cardiotoxicity in AML and be able to calculate more precise stratification scores.

#### 4.4.4. Pericardial Toxicity

Many of the novel agents used in the treatment of AML in adults have been attested to trigger pericardial toxicity. A study by Janus et al., 2022, discussed the different therapeutic regimens that can cause pericardial events, including pericarditis and pericardial effusion, and found out that they occurred less often in patients treated with cladribine (0.3%), fludarabine (0.4%), Venetoclax (0.3%), enasidenib (0.3%), and ivosidenib (0.3%) compared to those treated with Cytarabine (0.9%), whereas tamponade arose significantly less often in patients receiving cladribine (0.1%), fludarabine (0.4%), enasidenib (0.1%), ivosidenib (0.1%), and venetoclax (0.1%) compared to those receiving cytarabine (0.7%) [48].

#### 4.4.5. Myocarditis [48]

Myocarditis is defined as “multifocal inflammatory cell infiltrates with overt cardiomyocyte loss by light microscopy” with a pathohistological diagnosis. There are criteria for clinical diagnosis that can include high-sensitivity cardiac troponin elevation and diagnostic MRI for myocarditis or specific clinical manifestations [47]. In the cohort by Janus et al., 2022, only 145 myocarditis events were reported during the study period [48] and, generally, anthracyclines and decitabine are known to be triggering factors for acute myocarditis.

### 4.5. Arrythmias

#### 4.5.1. Atrial Fibrillation (AFib)

A-Fib is the most common cardiac arrhythmia, with an overall prevalence of 1–2% in the US. Advancing age is a prominent risk factor for both AFib and AML [49]. Regarding the toxicity by AML-specific regimens, when analyzed for AFib as a percentage of all reported AEs for a particular drug, venetoclax was reported to have a prevalence of AFib at 1.6% and 29% compared to its associated cardiac AEs alone [37]; treatment with azacitidine was correlated with atrial fibrillation (3% to 5%) [14]. Atrial fibrillation is also strongly associated with midostaurin use in AML patients [25,26].

#### 4.5.2. Sinus Bradycardia

Cytarabine-induced symptomatic sinus bradycardia has been substantiated in the recent literature, although it is a not a well-known AE associated with its use [50,51,52]. In the retrospective cohort study by Romani et al., 2009, an incidence of 2.8% was reported in patients receiving intermediate- or high-dose cytarabine [52].

#### 4.5.3. QTc Prolongation

QT prolongation in cardio-oncology is defined as a corrected QT interval using the Fridericia correction (QTcF = QT/3√RR) with a duration over 500 ms, which requires immediate cessation of the therapeutic regimen given [47].

Boluda et al.’s 2023 study included a detailed analysis of the occurrence of QT prolongation in real-time in AML patients. The frequency of QTcF prolongation in a clinical trial with FLT3 inhibitor quizartinib (QuANTUM-First) was reported as 34%. Furthermore, 21% of the patients included in front-line clinical trials (the vast majority without FLT3 inhibitors) developed QTcF prolongation [27]. Other targeted therapies related to QTcF prolongation are midostaurin [24], gilteritinib [31], glasdegib [33], ivosidenib [35], and olutasidenib (8%) [29].

### 4.6. Anthracycline-Induced Cardiotoxicity

There are two main categories of anthracycline-induced cardiotoxicity: acute and chronic. The latter is more common and happens in a dose-dependent manner. Usually not dose-dependent, acute cardiotoxicity might manifest as myocarditis, arrhythmia, or sudden heart failure. Chronic cardiotoxicity can manifest clinically in a variety of ways, from asymptomatic LVEF decrease to heart failure. Cardiotoxicity, which is commonly defined as a decline in the LVEF of >10% to a final LVEF of <50%, is often observed after the use of anthracyclines. Nevertheless, the published research differs in its precise cutoff values for LVEF reduction. While they have occasionally been utilized, additional criteria including reduced left fractional shortening, aberrant wall motion, global longitudinal strain, and diastolic dysfunction have not yet been included in the conventional assessment and classification of anthracycline-related cardiotoxicity [10,11]. There have been numerous terminologies proposed for cancer therapy-related cardiovascular toxicity (CTR-CVT), causing inconsistencies in diagnosis and management. To address this, the ESC 2022 guideline provides consensus definitions for cardiomyopathy, heart failure (HF), myocarditis, vascular toxicities, hypertension, cardiac arrhythmias, and QT interval prolongation. It recommends using “cancer therapy-related cardiac dysfunction (CTRCD)” for cardiac injury, cardiomyopathy, and HF to encompass the wide range of presentations and link them to various cancer treatments, including chemotherapy and radiation. Other CTR-CVT definitions, such as those for pericardial and valvular heart diseases, align with general cardiology standards. The guideline also includes detailed criteria for assessing the severity of and recovery from conditions like myocarditis [47].

## 5. Conclusions

Thus, based on the scoping review conducted, we concluded that the correlation between the pharmaceutical approach provided to AML patients and the incidence of cardiotoxicity is of high importance. It is understandable that being familiar with all the available treatment options available and every potential adverse effect is impossible before the initiation of the therapy. However, hematologists and, in general, physicians should try to be updated with the most recent information released to improve the quality of life of their patients and minimize the risk of additional complications. Correct communication and collaboration between hematologists and cardiologists is of paramount importance and a huge advantage that can yield promising results, utilizing both cardiology imaging techniques, such as cardiac ultrasound, which is an easy and cost-effective means of instant diagnosis and staging the suspected complication, and serum cardiac biomarkers, according to current cardio-oncology guidelines. Furthermore, the management of such complications must be the purpose of future studies, especially clinical and preclinical studies, that address disease-specific recommendations in the management of AML patients rather than drug-specific recommendations, due to the specificity of those patients. However, from this research, the authors recommend the immediate discontinuation of the pharmaceutical regimen causing the cardiovascular complication occurring and change this to another treatment option. In the particular case of anthracycline cardiotoxicity, which has been thoroughly studied, it is known that such complications are dose-dependent and possible dose down-regulation is an option, but a change to the regimen should be considered.

## Figures and Tables

**Figure 1 cancers-16-02474-f001:**
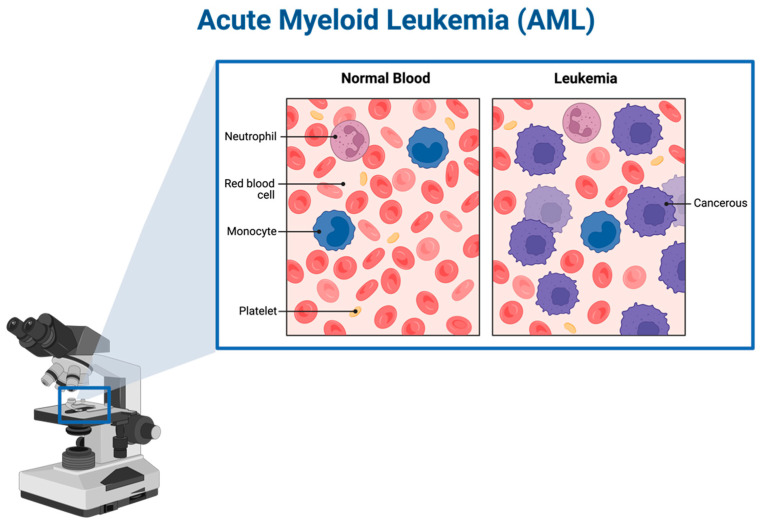
AML under the optical microscope, created using BioRender [accessed on 20 May 2024). Available online: https://app.biorender.com/illustrations/65ff040d4603630b1370a481].

**Figure 2 cancers-16-02474-f002:**
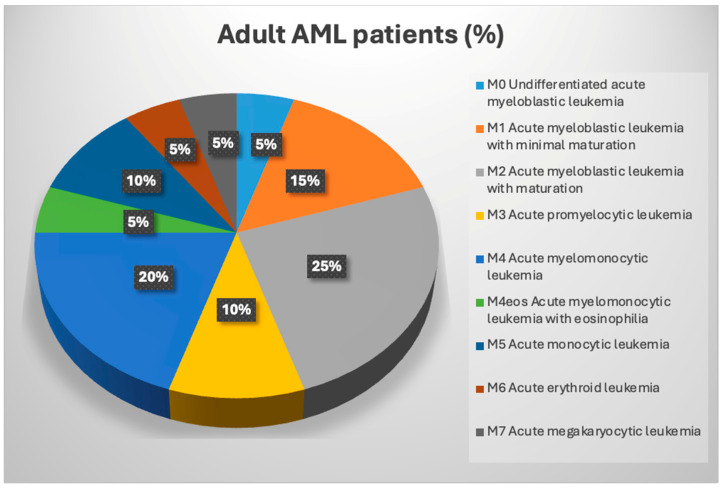
AML subtypes and rates.

**Table 1 cancers-16-02474-t001:** WHO classification of AML and related neoplasms (‘Acute Myeloid Leukemia (AML) Staging: FAB and WHO Classifications for Acute Myeloid Leukemia’, 2023).

AML Categorisation	
**By Genetic Abnormalities**	**By Differentiation**
*RUNX1::RUNX1T1* fusion	Minimal differentiation
*CBFB::MYH11* fusion	Without maturation
*DEK::NUP214* fusion	With maturation
*RBM15::MRTFA* fusion	Acute basophilic leukemia
*BCR::ABL1* fusion	Acute myelomonocytic leukemia
K *MT2A* rearrangement	Acute monocytic leukemia
*MECOM* rearrangement	Acute erythroid leukemia
*NUP98* rearrangement	Acute megakaryoblastic leukemia
*NPM1* mutation	-
*CEBPA* mutation	-
myelodysplasia-related	-

**Table 2 cancers-16-02474-t002:** Pharmaceutical regimens currently used in the treatment of AML in adults.

Regimen	Mechanism of Action	Indication in AML Treatment	Adverse Effects (Common)	Cardiovascular Toxicity (Reported)
Systematic Chemotherapy Drugs				
**“7 + 3” (cytarabine + anthracycline)**		Induction therapy	Allergic reactions (anaphylaxis, rash) Fever Severe cough Photosensitivity Skin and nail hyperpigmentation Hoarseness of voice Bone marrow suppression, low blood cell counts Mucositis Fatigue Joint pain Persistent diarrhea	Anthracycline-induced cardiotoxicity (acute HF, arrhythmia, or myocarditis, and anthracycline-related left ventricular dysfunction [ARLVD])
**Antracyclines (daunorubicin, idarubicin, mitoxantrone)**	**Inhibition of DNA synthesis and DNA topoisomerase II**			
**Hypomethylating agents (HMA; azacitidine and decitabine)**	**Inhibition of DNA methylotransferase**	Maintenance following intensive chemotherapy HSCT ineligible	Pancytopenia Hepatotoxicity Acute renal injury	Chest pain Atrial fibrillation (3–5%) Congestive heart failure (3–11%) Cardiomyopathy (<5%) Hyper/hypotension
**Cytarabine**	**Antimetabolic agent and inhibition of DNA polymerase**	Induction therapy	Gastrointestinal disturbances Stomatitis Conjunctivitis Reversible hepatic enzyme elevation Dermatitis	Anthracycline-associated cardiomyopathy, pericarditis, bradycardia (2.8%), angina pectoris
**CPX-351 (daunorubicin and cytarabine)**		Secondary AML Frontline	GI disturbances (constipation, diarrhea, nausea, vomiting) Cytopenias Allergic reactions Ototoxicity	Anthracycline-associated cardiomyopathy Edema Arrhythmia Hypo/hypertension Chest pain
**Targeted Therapy**				
**GO (gemtuzumab ozogamicin)**	**Anti-CD33 monoclonal antibody**	Favorable/intermediate/unknown cytogenetics Patients with CD33+-expressing leukemic blasts Frontline with intensive chemotherapy	Neutropenia and thrombocytopenia Hepatotoxicity and portal fibrosis	Infusion-related hypotension (5%) Acute LV dysfunction
**Midostaurin**	**Multi-kinase inhibitor against FLT3-ITD and FLT3-TKD**	Newly diagnosed FLT3 ITD/TKD-mutated AML Frontline with intensive chemotherapy	Pancytopenia Infection, mucositis or stomatitis, pneumonitis or pulmonary infiltrates Diarrhea, nausea, increased alanine aminotransferase, hyperbilirubinemia Hypokalemia, hyponatremia, hypophosphatemia, hypocalcemia Pain, rash, fatigue	QTc prolongation, heart failure, atrial fibrillation, hypotension and pericardial disease
Quizartinib	FLT3-inhibitor	Relapsed/refractory FLT3-mutated AML (only in Japan)	Prolonged cytopenias	QTc prolongation (34%)
Enasidenib	Inhibition of mutant IDH2 enzyme	Mutant IDH2 R/R AML	Cytopenias GI symptoms (nausea, vomiting, decreased appetite) Blood bilirubin increased Differentiation syndrome	N/A
Gilteritinib	FLT3-inhibitor	Monotherapy for R/R FLT3-TKD or ITD AML	Pancytopenia Hepatotoxicity Pyrexia Diarrhea, constipation Vomiting Hypokalemia Pneumonia Peripheral edema	QTc prolongation (4.9%) Risk of heart failure development (4%) Edema Hypotension Pericardial effusion Myocarditis Pericarditis
Glasdegib	Inhibitor of the hedgehog signaling pathway by binding to the smoothened (SMO) receptor	Newly diagnosed AML in patients aged ≥75 years or those who have comorbidities that preclude use of intensive induction chemotherapy (+cytarabine)	Febrile neutropenia, nausea, decreased appetite, fatigue, constipation, dysgeusia, muscle spasms, dizziness, and vomiting	QTc prolongation Peripheral edema
Ivosidenib	Selective inhibitor of mutant IDH1 enzyme	Frontline monotherapy for ND IDH1-AML if ≥75 years or unfit for IC Frontline IVO/AZA for ND IDH1-AML if ≥75 years or unfit for IC Monotherapy for R/R IDH1-AML	Differentiation syndrome (DS) and leukocytosis	QTc prolongation (18–25%) Edema Hypotension Chest pain Ventricular fibrillation (<1%)
Olutasidenib	Inhibitor of mutant IDH1 enzyme	Monotherapy for IDH1-AML	Cytopenias Acute liver injury and failure (ALT, AST, gGT) GI side effects Pneumonia Hypokalemia DS Tumor lysis syndrome	QTc prolongation (8%)
Venetoclax	Anti-BCL2 monoclonal antibody	Elderly/unfit Frontline with azacitidine, decitabine	Pancytopenia (>10%) Nausea, constipation, diarrhea, vomiting (>30%) Hypokalemia Pneumonia (38%) Sepsis (6%) Tumor lysis syndrome (1%)	Atrial fibrillation (5%)

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
