# Peer review of "Cardiotoxicity in Acute Myeloid Leukemia in Adults: A Scoping Study"

_cancers, 2024, doi:10.3390/cancers16132474_

Round 1

Reviewer 1 Report

Comments and Suggestions for Authors

Content suggestions:

1.         Could the Authors propose the management of patients affected by cardiovascular complications developed due to the treatment of AML ?

Formal suggestions:

1.         I would like to kindly ask the Authors to remove the significant part of the first two chapters discussing the general facts about AML.

Author Response

  1. Comment: Could the Authors propose the management of patients affected by cardiovascular complications developed due to the treatment of AML?

Response: Dear Reviewer, thank you for your comment. The essence of our paper was to create a scoping review regarding the therapeutic treatments of AML and the possible adverse effects regarding cardiotoxicity. We appreciate your suggestion but in order for the paper to remain targeted on its original question – which is to provide easier accessibility to physicians regarding the adverse effects of AML medications, the paragraph suggested wouldn’t be appropriate to be intergraded.

  1. Comment: I would like to kindly ask the Authors to remove the significant part of the first two chapters discussing the general facts about AML.

Response: Dear Reviewer, thank you for your comment. Could you specify on which parts of the Introduction you want us to adjust. We believe that the content of the Introduction is crucial in order for young physicians and residents who come in contact with such topic for the first time to get familiarised with the content.

Reviewer 2 Report

Comments and Suggestions for Authors

The article "CARDIOTOXICITY IN ACUTE MYELOID LEUKEMIA IN ADULTS: A SCOPING STUD" is well written and structured in dreams of its section and passage.

The only point that could be better developed is regarding data on cardiotoxicity monitoring, which could be added, as for example reported in the article by Cantoni et al (Cantoni V, Green R, Assante R, et al. Prevalence of cancer therapy cardiotoxicity as assessed by imaging procedures: A scoping review. Cancer Med. 2023; 12: 11396-11407. doi:10.1002/cam4.5854).

I suggest this point because the authors mentioned this point in the last sentence of the "Conclusion" section.

Author Response

Comment: The only point that could be better developed is regarding data on cardiotoxicity monitoring, which could be added, as for example reported in the article by Cantoni et al (Cantoni V, Green R, Assante R, et al. Prevalence of cancer therapy cardiotoxicity as assessed by imaging procedures: A scoping review. Cancer Med. 2023; 12:11396-11407. doi:10.1002/cam4.5854). I suggest this point because the authors mentioned this point in the last sentence of the "Conclusion" section.

Response: Dear Reviewer, we want to thank you for your comment. We appreciate your suggestion but since the paper is mainly focused on the pharmaceutical treatments, we think that it would rather confuse the reader as it was not included in depth on the main text. In addition, we highly recommend the use of the cardiology society guidelines, as mentioned in the main text, regarding diagnosis and monitoring of the cardiotoxicity-related adverse effects.

Reviewer 3 Report

Comments and Suggestions for Authors

In this study the Authors focused on the analysis of the 

consequences of AML for cardiac function.

This comprehensive manuscript attracts attension 

to very important issue and therefore it can be published 

in Cancers. There are only several small conserns. 

1. Abstract. Please, describe main manifestations of 

cardiotoxicity associated with AML.

2. Please, polish writing style in Abstract and throught

the text.

3. Please, write a small chapter describing the main 

strategies of cardiotoxicity mitigation and prevention.The

Chapter can include the description of the best cardiopfotecors, recommendations of moderate exercises, walking, psychotherapy, cognitive therapy, art

and zootherapy and other. 

Comments on the Quality of English Language

Minor writing style correction is needed.

Author Response

  1. Comment: Please, describe main manifestations of  cardiotoxicity associated with AML. And 2. Please, polish writing style in Abstract and throught the text.

Response: Dear Reviewer, we want to thank you for your comment. Please find the corrections highlighted in the Abstract and Main Text.

  1. Comment: Please, write a small chapter describing the main strategies of cardiotoxicity mitigation and prevention. The Chapter can include the description of the best cardiopfotecors, recommendations of moderate exercises, walking, psychotherapy, cognitive therapy, art and zootherapy and other. 

Response: Dear Reviewer, thank you for your comment. As this type of content was not initially investigated it will interfere with the structure of the scoping review.